METHODS AND RESOURCES

# High-resolution 3D imaging and topological mapping of the lymph node conduit system

**Inken D. Kelch**[1,2]*, **Gib Bogle**[1,3], **Gregory B. Sands**[3], **Anthony R. J. Phillips**[1,2,4], **Ian J. LeGrice**[3,5], **P. Rod Dunbar**[1,2]*

**1** Maurice Wilkins Centre for Molecular Biodiscovery, University of Auckland, Auckland, New Zealand,
**2** School of Biological Sciences, Faculty of Science, University of Auckland, Auckland, New Zealand,
**3** Auckland Bioengineering Institute, University of Auckland, Auckland, New Zealand, **4** Department of
Surgery, School of Medicine, Faculty of Medical and Health Sciences, University of Auckland, Auckland, New
Zealand, **5** Department of Physiology, School of Medical Sciences, Faculty of Medical and Health Sciences,
University of Auckland, Auckland, New Zealand

* i.kelch@auckland.ac.nz (IDK); r.dunbar@auckland.ac.nz (PRD)

**Data Availability Statement:** Computational tools developed in this study are available on GITHUB (https://github.com/gibbogle/vessel-tools.git). All relevant data are within the paper and its Supporting Information files.

## Abstract

The conduit network is a hallmark of lymph node microanatomy, but lack of suitable imaging technology has prevented comprehensive investigation of its topology. We employed an extended-volume imaging system to capture the conduit network of an entire murine lymph node (comprising over 280,000 segments). The extensive 3D images provide a comprehensive overview of the regions supplied by conduits, including perivascular sleeves and distinctive "follicular reservoirs" within B cell follicles, surrounding follicular dendritic cells. A 3D topology map of conduits within the T-cell zone showed homogeneous branching, but conduit density was significantly higher in the superficial T-cell zone compared with the deep zone, where distances between segments are sufficient for T cells to lose contact with fibroblastic reticular cells. This topological mapping of the conduit anatomy can now aid modeling of its roles in lymph node function, as we demonstrate by simulating T-cell motility in the different T-cell zones.

## Introduction

Sophisticated immune responses are organized within the highly structured microanatomy of lymph nodes (LNs) where stromal cell networks support the circulation, maintenance, and interaction of highly motile hematopoietic cell types on their continuous quest for cognate antigen [1–3]. A key feature of the LN organization is the mesh-like network of fibroblastic reticular cells (FRCs) spanning the LN paracortex, the main homing zone for T cells [4,5]. FRCs organize LN microenvironments and control T-cell life in many ways by providing survival signals, aiding migration, and restricting T-cell activation [6,7]. They express the chemokines C-C motif chemokine ligand 19 (CCL19) and C-C motif chemokine ligand 21 (CCL21), important cues for motility, compartmentalization, and retention of C-C chemokine receptor type 7 (CCR7)-expressing T cells, B cells, and dendritic cells (DCs) [3,8,9]. In a similar fashion, FRCs appear to be involved in B cell homeostasis, by providing the B cell survival factor BAFF

**Funding:** Funding was provided by the Maurice Wilkins Centre for Molecular Biodiscovery (IK & PRD). GB, GBS, IJL, and AP received no specific funding for this work. The funders had no role in study design, data collection and analysis, decision to publish, or preparation of the manuscript.

**Competing interests:** The authors have declared that no competing interests exist.

**Abbreviations:** BAFF, B cell activating factor; CCL19, C-C motif chemokine ligand 19; CCL21, C-C motif chemokine ligand 21; CCR7, C-C chemokine receptor type 7; CLEC-2, C-type lectin-like receptor 2; CS, cortical sinus; CXCL13, chemokine (C-X-C motif) ligand 13; DAPI, 4',6-diamidino-2-phenylindole; DC, dendritic cell; EVIS, extended-volume imaging system; FDC, follicular dendritic cell; FRC, fibroblastic reticular cell; HEV, high endothelial venule; IgM, immunoglobulin M; IL-7, interleukin 7; LN, lymph node; LYVE-1, lymphatic vessel endothelial receptor 1; MC, medullary cord; SCS, subcapsular sinus; SD, standard deviation; TCZ, T-cell zone; WGA, wheat germ agglutinin.

(B cell activating factor) and contributing to chemokine (C-X-C motif) ligand 13 (CXCL13) expression [10,11]. LN expansion during immune stimulation is mediated by FRCs in synergy with DCs, which can trigger FRC stretching via interaction of C-type lectin-like receptor 2 (CLEC-2) with podoplanin [12,13]. FRC destruction is part of the pathology of several devastating viral diseases and directly affects the number and functionality of T cells [2,7,14]. FRC networks also appear in tertiary lymphoid structures at sites of chronic inflammation underlining their central importance to immunobiology [15,16].

Remarkably, FRCs construct a piping system that rapidly conducts incoming lymphatic fluid, including tissue-derived antigens across the LN cortex [17–19]. This conduit system consists of interconnected "micro vessels" built of a central core of collagen fibers surrounded by a layer of microfibrils and a basement membrane enwrapped by FRCs and channels lymph from the subcapsular sinus (SCS) to inner LN compartments [17–20]. Access to the conduit system appears restricted to molecules defined by low molecular weight (<70 kDa), which was originally attributed to the structural properties of the core of collagen fibers [18,21]. However, recent evidence suggest that the 70 kDa "filter" is established by endothelial cells in the SCS [20] and that conduits are in fact able to carry larger molecules, such as antibodies, and even occasionally pass small virions [22,23]. In particular, inflammatory soluble mediators and cytokines can be shuttled directly to high endothelial venules (HEVs), specialized vessels for lymphocyte entry that are surrounded by perivascular "sleeves" formed by FRCs [5,24–26]. Intriguingly, the conduit network persists even if FRCs are temporarily lost, suggesting that it possesses structural integrity, while depending on FRCs for remodeling [10]. Many questions remain concerning the heterogeneity of FRC populations, the exact mechanisms by which they regulate immunity and the advantages of FRC-guided migration of T cells in a 3D space [7,27]. Our understanding of the structure of the conduit network remains limited because of the technical difficulty of capturing these delicate network structures within large tissue volumes [28]. Previous approaches to studying the FRC network globally within LNs have relied on in silico computer models with predefined network properties [29–31], based on information from confocal images on a small scale [32,33]. Large-scale 3D imaging of entire networks has to date been hindered by the limitations of tissue penetration in standard microscopy and restrictions in resolution of large-scale imaging techniques [34,35]. An additional complexity is that moving from small-scale measurements in 2D to large-scale measurements in 3D requires specialized nontrivial algorithms that often require custom computation by the operating lab to fit a particular purpose [35].

To provide a comprehensive picture of the LN conduit network we used a unique confocal block-face imaging system referred to as EVIS (extended-volume imaging system) [36,37] and captured the conduit and blood vessel system of an entire murine LN. EVIS is a custom-built imaging platform that allows large tissue volumes (extending over several milimeters) to be imaged at subcellular resolution (down to 0.5 μm) by combining confocal microscopy with repeated rounds of precise sample trimming and ultimately integrates overlapping image stacks into seamless 3D images. It is therefore particularly suited for imaging the fine conduit network across a tissue volume as big as a whole lymph node. From the acquired seamless 3D images, we extracted a continuous topology map of the conduits in the T-cell zone (TCZ) and quantified the network structure with the help of custom image-processing tools. The obtained topology map permitted the assessment of 3D network parameters at unprecedented scale and served as a realistic template for in silico simulations of T-cell motility. Our measurements revealed significant differences in conduit segment density between the deep and superficial TCZs, making it likely that T cells in the deep zone lose contract with the FRC network more frequently. We were surprised to find distinctive tracer accumulations in the B cell follicles, and we visualized the intriguing organization of the conduit-supplied spaces surrounding

FDCs with new clarity. Our topology map provides a unique reality-based road map of the intricate 3D organization of the LN conduits that can be incorporated into the increasingly sophisticated theoretical models seeking to understand and predict complex immune processes within LNs [38].

## Results

### Extensive 3D imagery permits volume views of the continuous conduit network

Previously, studies of the LN conduit system have relied on microscopic images with limited depth information. By performing EVIS imaging at a voxel resolution of 1 μm, we were able to capture a popliteal LN sized 850 × 750 × 900 μm in its entirety. Organ-wide anterograde labeling of the lymphatics and blood vessels was achieved by injecting wheat germ agglutinin (WGA) conjugated to different fluorophores into the footpad and the supplying blood vessel, respectively. The resulting 3D image permits detailed insights into the overall LN anatomy (Fig 1). As a 38 kDa molecular tracer, WGA recapitulates the routes of lymph-borne molecules <70 kDa through the LN. Strong WGA labeling can be seen in the SCS and the medulla, thereby fully enclosing the LN. By virtually cropping the 3D volume (Fig 1A), views of the interior organization (Fig 1B) and the dense network of blood vessels running through the LN are revealed (Fig 1C). The conduit network is most structured in the central TCZ (Fig 1D) and is sparse in the B cell follicles, with only a few channels running beside any one follicle (Fig 1E). The medulla is richly filled with WGA, providing a high staining intensity in the lymphatic sinuses, yet medullary cords (MC), strands of parenchymal tissue that extend into the medullary space and are densely packed with cells [5,39], are clearly distinguishable and contain at least one central blood vessel (Fig 1F).

Within the TCZ, the conduit network appears most dense in the superficial and interfollicular zones, whereas a sparser network structure becomes apparent within its center (Fig 1D, 1G–1J), consistent with previous definitions distinguishing the deep TCZ from surrounding regions [40]. Particularly strong staining could also be observed around HEVs and smaller blood vessels, which both appear surrounded by a sleeve contiguous with the conduit network (Fig 1G–1J). However, intraluminal staining of blood vessels with lymph-derived WGA was not observed (Fig 1G–1J, S2 Fig). Cortical sinuses (CS) [41] also display strong labeling, but can be distinguished from blood vessels by lack of blood vessel-specific WGA-staining (Fig 1J) and their continuity with the medullary sinuses, a feature that becomes evident in animations of the 3D data set (S1 and S2 Videos). Interestingly, although it was previously reported that conduits are primarily focused on HEVs, we observed in our 3D images that conduits frequently terminate on CS, which are often located in close proximity to blood vessels (S1 Fig). Examining tissue sections using conventional immunofluorescence microscopy confirmed that conduits connect to CS made up of lymphatic vessel endothelial receptor 1 (LYVE-1) expressing lymphatic endothelial cells (S1 Fig). Together, these images demonstrate that the conduit system connects the SCS with CS that drain into the medulla, as well as the perivascular sleeves surrounding blood vessels including HEVs, thereby providing a continuous piping system for incoming lymphatic fluid (S1 and S2 Videos).

### Quantification of the conduit network topology

The availability of an extensive 3D volume image of the continuous LN conduit network permits quantification of its network statistics at unprecedented scale and provides an exciting opportunity for the realistic modeling of T-cell motility. We previously imaged and quantified

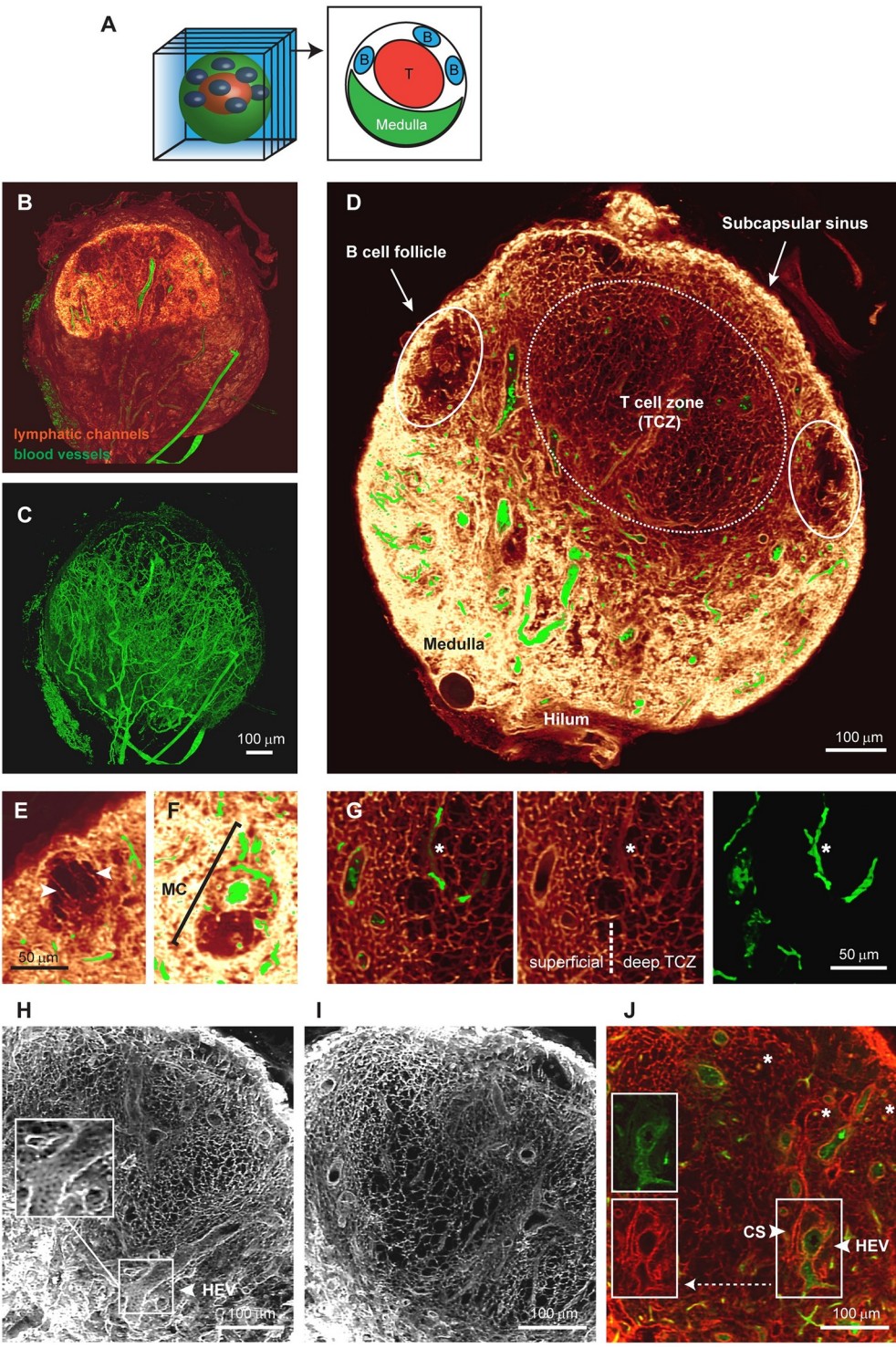

**Fig 1. Detailed 3D images of the conduit network in a whole LN.** EVIS imaging of an entire popliteal LN generated a 3D volume image of which interior slices can be viewed individually (A). Three-dimensional image reconstruction of the entire LN shows (B) lymphatic channels filled with the tracer molecule WGA (red glow) together with dextran-labeled blood vessels (green), or the blood vasculature alone (C). An interior view of 20-μm thick optical sections (D—I) and a 1 μm slice (J) permits detailed insights into the LN architecture. A cross-section of the LN displays the location of cell-specific zones (D), and close-ups reveal anatomical details, including the arrangement of long conduits descending from the SCS at the edges of a B cell follicle (arrowheads, E), an MC with a central blood vessel situated amongst the WGA-filled sinuses of the medulla (F), and the transition from dense to sparse conduit networks in the

superficial to the deep TCZ (G). The conduit network forms a highly organized grid within the TCZ (white, H and I; red, J) interspersed with CS (arrowhead, panel J) and blood vessels (green), including HEVs, which are closely surrounded by cells displaying a cobblestone-like morphology (arrowhead, panels H and J). Besides larger blood vessels, small blood vessels are frequently enclosed by conduits (asterisks, G and J). Image rendering was performed in Voxx (A—I) and ImageJ (J). See also S1 Fig, S1 and S2 Videos. CS, cortical sinus; EVIS, extended-volume imaging system; HEV, high endothelial venule; LN, lymph node; MC, medullary cord; SCS, subcapsular sinus; TCZ, T-cell zone; WGA, wheat germ agglutinin.

the blood vessel system of a mesenteric LN using a set of custom-developed image-processing and analysis tools [36] and now applied these tools to perform large-scale 3D analysis on the conduit network. The image processing consists of a number of steps, including thresholding and skeletonization, which transform the pixel-based image data into a 3D topology map. The topology map describes the network as a system of connected tubes and enables a direct read-out of network parameters (Fig 2). In order to study the network topology of the conduit system in the central paracortical TCZ and its implications for T-cell biology, the extraction procedure was optimized to best capture the network in this region (Fig 2A–2C). A limitation of this process was posed by the occurrence of continuous tracer-labeled spaces fully enclosing large blood vessels, such as HEVs (S2 Fig, S3 Video), identified previously as perivascular sleeves [5]. This feature of the conduit network provided an obstacle for the skeletonization process (S3 Fig) and required us to adapt our image-processing strategy. We overcame this problem by utilizing the co-stained blood vessels and subtracting the segmented blood vessel image data from the conduit image. In our previous study [36], we found blood vessels in the LN typically have diameters between 4 and 87 μm, whereas diameters of conduits are reported to lie in the range of 1 to 2 μm [17,19–21]. By removing the blood vasculature from the conduit data, vessels of the size of blood vessels could be effectively excluded (Fig 2D–2F). The resulting "clean" conduit network contained 282,716 segments with a mean diameter of 2.9 μm and an average length of 6.5 μm (Figs 2G, 2H and 3H and 3I). Within the TCZ conduit network, spanning a volume of about 0.079 mm$^3$, the conduit segments had a combined length of 1.84 m and a density of $3.54 \cdot 10^6$ segments mm$^{-3}$ (Fig 3F and 3J). To obtain a measure of spacing in the network, we applied an algorithm that measures the distance to the nearest conduit segment starting from a regular fine grid of points located in the LN volume [36]. This calculation revealed that the majority of locations in the LN TCZ lie within a very short distance of the nearest conduit (<4 μm, 90.9%; Fig 2I). Overall, the conduit network displayed an even branching pattern, with the majority of branching points representing bifurcations and branching angles centered around 120˚ (Fig 2J and 2K). The segment orientation had no observable bias in direction (Fig 2L).

We then tested how this network topology would predict the migration of T cells in 3D, when simulated T cells are restricted to migrating along the network segments, as if in continuous contact with FRCs. We simulated the paths of a large number of cells on the extracted conduit network to calculate the coefficient of motility ($C_m$) as an index of dispersal rate in 3D space, representing the rate at which T cells can scan a volume of paracortex for the presence of cognate antigen. In these simulations, we used values of mean speed in the range typically measured by intravital microscopy [42–46]. The average displacement of cells (Fig 2M) at T = 60 min was used to calculate $C_m$, (Fig 2N), and the correlation we generated between mean speed and $C_m$ was broadly consistent with values previously measured in vivo [43]. The average displacement and the corresponding $C_m$ values are not significantly increased when blood vessels are included in the analysis (Fig 2M and 2N), but cell tracks show a slight variation due to the availability of the blood vessels and the sheaths that often surround them as additional migration paths (Fig 2O).

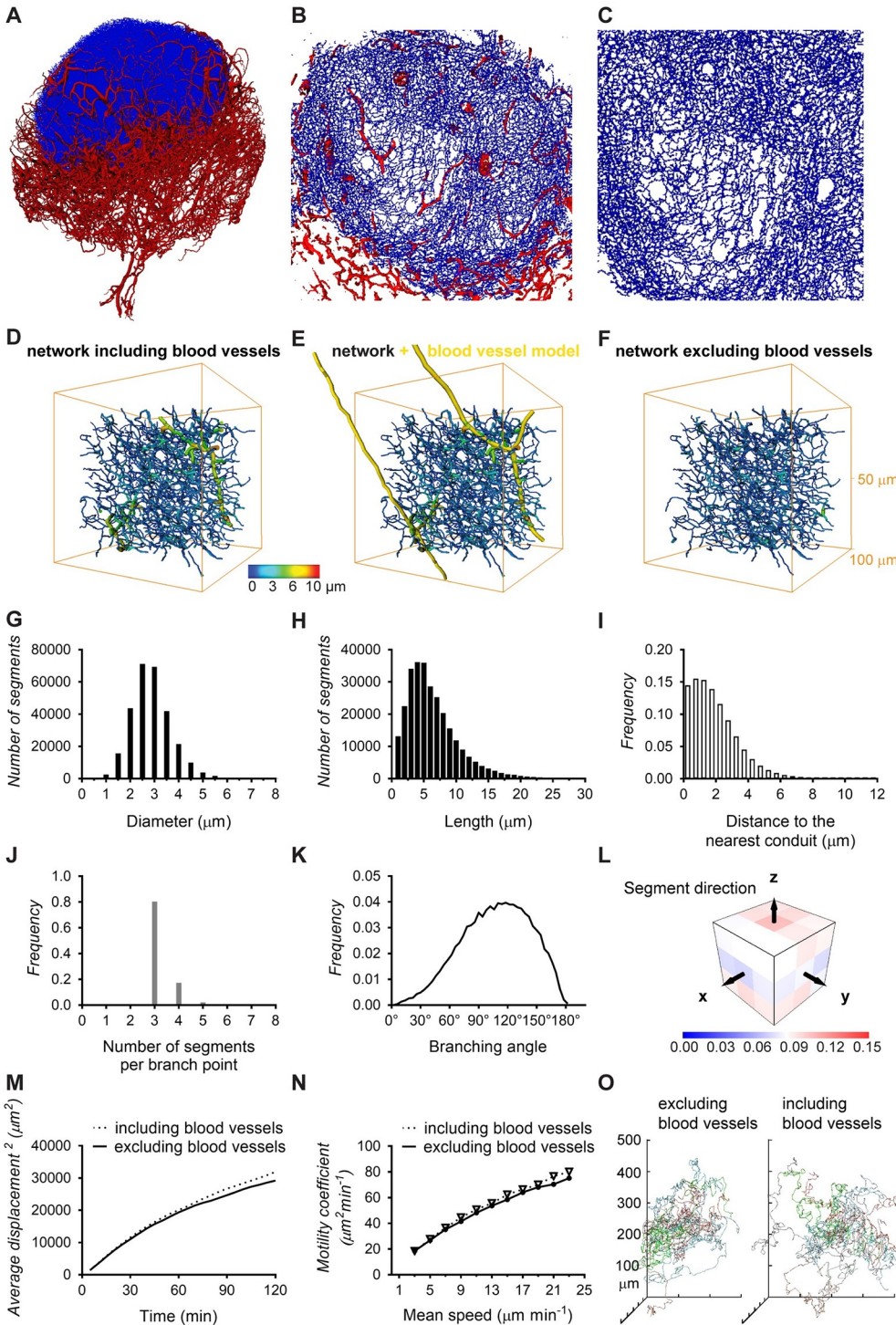

**Fig 2. Network topology of the LN conduits in the TCZ.** Deploying custom-developed image-processing tools, a description of the conduit network in terms of nodes and links was generated from the 3D image data and used to estimate network parameters. Three-dimensional projections of the blood vasculature (red) and the conduit network in the TCZ (blue) as a whole (A) and magnified views of the TCZ (B, C) expose the complexity and the high level of detail in this data set. The full conduit data include large segments with diameters over 5 μm (D), but these represent blood vessels as the overlay with the blood vessel model (yellow) indicates (E). A blood vessel-free conduit network (F) was obtained by removing the majority of blood vessels from the 3D image prior to the network extraction in a semi-automated process. This TCZ conduit network excluding blood vessels was employed to calculate the distribution of

segment diameters (G), lengths (H), the branching pattern (J), branching angles (K), and segment orientation (L), and the full data set including blood vessels was used to calculate the minimum distance to the nearest conduit (I). Simulation of T-cell motility utilizing these conduit data provides the cell displacement at a mean speed of 13 μm min$^{-1}$ (M), motility coefficients for different speeds (N), and a spider plot representation of migration paths in a network with and without blood vessels (O). Values for each data point can be found in S1 Data. See also S2 and S3 Figs, and S3 Video. LN, lymph node; TCZ, T-cell zone.

## Topology differences in the deep and superficial TCZ

It was evident in the 3D conduit image and the topology map that the conduit network in the TCZ is not homogeneous but displays different densities in the superficial and deep zone (Fig 3), concordant with previous descriptions [40]. After coloring regions based on their segment density, it is possible to visually distinguish the deep TCZ, containing a rather open mesh, from the superficial zone, which is characterized by a dense network of conduits and fully encloses the spherical central T-cell region (Fig 3A and 3B). In a different approach to visualizing the variable spacing in the network, distances to the nearest conduit segment were measured in 3D, and locations further than 8 μm from any conduit were displayed in red (Fig 3C and 3D). An accumulation of red voids is located centrally in the deep TCZ, while they were absent from superficial locations. To quantify these regional differences, 5 spherical subregions with a diameter of 100 μm were selected from the superficial and deep TCZ each and examined using the topology toolset (Fig 3D). The deep TCZ contained significantly fewer vertices, segments, and a smaller conduit volume per region than the superficial zone, confirming visually observable differences in conduit density (Fig 3E–3G). Although the conduit diameters in both locations showed no measurable difference, individual segment lengths were considerably shorter in the superficial zone, yet the combined conduit length of all segments was longer than in the deep TCZ (Fig 3H–3J). In summary, the deep TCZ can be perceived as a stretched version of the conduit network in superficial areas. As a result, cells in the deep TCZ have a 50% greater mean distance to the nearest conduit segment (Fig 3L), reaching distances well beyond the cell diameter of a murine lymphocyte (2.5–3 μm) [47] and making it unlikely that cells in this region are in contact with a conduit segment at all times. In contrast, distances measured in the superficial zone would allow nearly continuous contact with the network (conduit distance <4 μm: 73.4% in the deep TCZ versus 96.8% in the superficial TCZ; conduit distance <6 μm: 92.2% deep versus 99.9% superficial; Fig 3K).

We then used our simulations of T-cell migration to predict motility coefficients separately in the superficial and deep zones, assuming that T cells remained in contact with the conduit network. The calculated 3D motility coefficients gradually increased with the speed of migration, but there were no significant differences in motility coefficients between the zones (Fig 3M).

With respect to potential specialized immune functions within the different TCZs, we noted differences in the distribution of proliferating T cells in 2D sections of resting LNs. Ki-67+ proliferating cells were often found in close proximity to the conduit network and seemed more frequent in the peripheral TCZ than the deep TCZ (S4 Fig), reinforcing the possibility that close cell contact (or the cues they provide) is important for T cells in the superficial TCZ.

## Conduit organization in B cell follicles

EVIS imaging of WGA-perfused LNs led to the intriguing observation of distinctive tracer accumulations inside B cell follicles. Compared to the dense conduit network in the TCZs, collagen-supported conduits are very rare in the B cell zones (Fig 4A and 4B), although a small number of conduits could often be visualized descending directly from the SCS, consistent

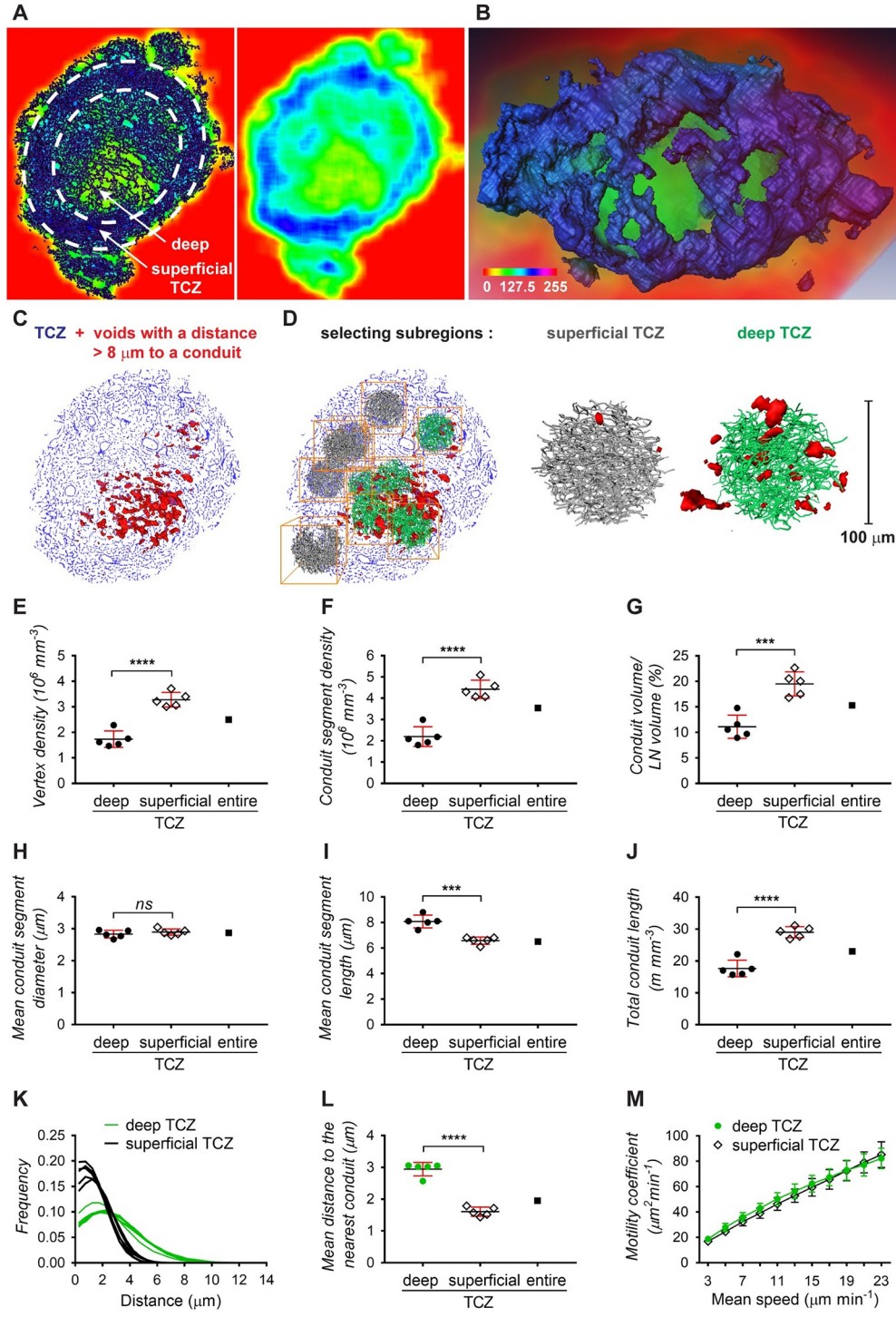

**Fig 3. Comparison of conduit network parameters in the deep and superficial TCZ.** Differences in conduit density between the deep and superficial TCZ can be visualized by averaging and color-coding pixel densities over small image volumes in a "moving average" display, shown as a rainbow spectrum (A, B). A cross-section of the moving average display exposes how dense regions in the periphery of the LN (blue) surround an inner region of lower conduit density (green), directly representing dense or sparse occurrence of conduit segments in the corresponding section of the conduit network image (dark blue, left panel), respectively (A). Volume rendering (B) of the entire TCZ using this approach shows the dense superficial zone (blue) enclosing a central region of sparse conduits (green). Alternatively, the TCZ conduit map was employed for calculating the distances to the nearest conduit, and voids with a distance of

over 8 μm were displayed in red, indicating larger distances within the deep TCZ as opposed to outer regions (C). From these 2 zones, 10 subregions were selected for comparative analysis (D), including the number of vertices (E), the number of conduit segments (F), the conduit volume (G), conduit segment diameters (H), conduit segment lengths (I), and the combined conduit length (J). The distributions of the minimum distances to the nearest conduit (K) and the average minimum distance (L) further exemplify the larger spacing within the deep TCZ. Simulation of T-cell motility predicts similar motility coefficients within the deep TCZ and the surrounding superficial zone (M). Data are from 1 experiment (each point represents one 100 μm subregion, $N = 10$) and plots show means ± SD. $****p < 0.0001$, $***p < 0.001$, ns = not significant, Student $t$ test. Values for each data point can be found in S1 Data. See also S4 Fig. LN, lymph node; TCZ, T-cell zone.

with channels previously referred to as follicular conduits [21]. However, we also observed distinctive WGA tracer accumulations within the B cell regions, appearing as discrete multilobular spaces reminiscent of "honeycombs" that are connected to the SCS and each other via follicular conduits (Fig 4A and 4B, S4 Video), occasionally aggregating into larger contiguous cavities spanning the entire B cell follicle. Hence these clusters appear as striking dense accumulations of WGA tracer within B cell zones that are otherwise relatively devoid of collagen-bearing conduits (Fig 4B–4D). To test how these WGA-filled spaces relate to the location of follicular DCs (FDCs), we used multicolor immunohistochemistry to identify FDCs in WGA-perfused LNs. The FDC marker CD21/CD35 co-localized with the observed deposits of WGA tracer (Fig 4C), confirming that the spaces we visualized surrounded and intercalated with FDCs deep within B cell follicles. Additional stains using collagen I to visualize conduit channels confirmed the transport of WGA through follicular conduits and deposition of WGA on FDCs (Fig 4D). Moreover, the arrangement and morphology of FDCs within the B cell follicle, as shown by co-staining with collagen I and a B cell marker, closely mirrors the location of the spaces typically filled by WGA (Fig 4E and 4F). High-resolution confocal image stacks revealed some diversity in the spaces where the WGA tracer accumulated within the follicles (Fig 4G and 4H, S5 Video). As well as the almost spherical structures of approximately 30 μm diameter that were brightly labeled, we noted weaker WGA tracer accumulation in adjacent honeycombed regions (Fig 4H, arrows), consistent with the various shapes of FDC clusters (Fig 4E and 4F). We also noted that the WGA signal inside B cell follicles was not as abundant in the 2D frozen sections (Fig 4C and 4D) compared with our 3D data (Fig 4A, 4B, 4G and 4H), suggesting that tracer may be washed off during frozen section preparation but is retained in the PFA-fixed and resin-embedded LNs we used for 3D imaging.

## Discussion

We set out to map the conduit network across an entire LN to enable measurements of its topology. Here we present extensive 3D imagery of the conduit channel system of a whole LN, permitting detailed insights into the conduit organization and its connectivity with blood and lymphatic vessels. We also provide a continuous reality-based computer representation of the TCZ conduits to enable large-scale quantification and downstream use in computer models of immune processes.

The conduit system has an intricate spatial relationship with the blood vessels and the lymphatic sinuses. Besides stabilizing the organ structure through their scaffold-like organization, conduits are thought to provide a short cut between incoming lymph and HEVs [18,19]. Although there is evidence that some molecules such as chemokines can gain access to the HEV lumen through the conduit network [18,19,25], this may depend on transport through endothelial cells by transcytosis [48]. We did not observe significant intravascular staining with WGA supplied into the conduits, implying that this 38 kDa molecule did not readily have access to the vascular lumen but was instead retained in a perivascular sleeve. Our 3D imagery

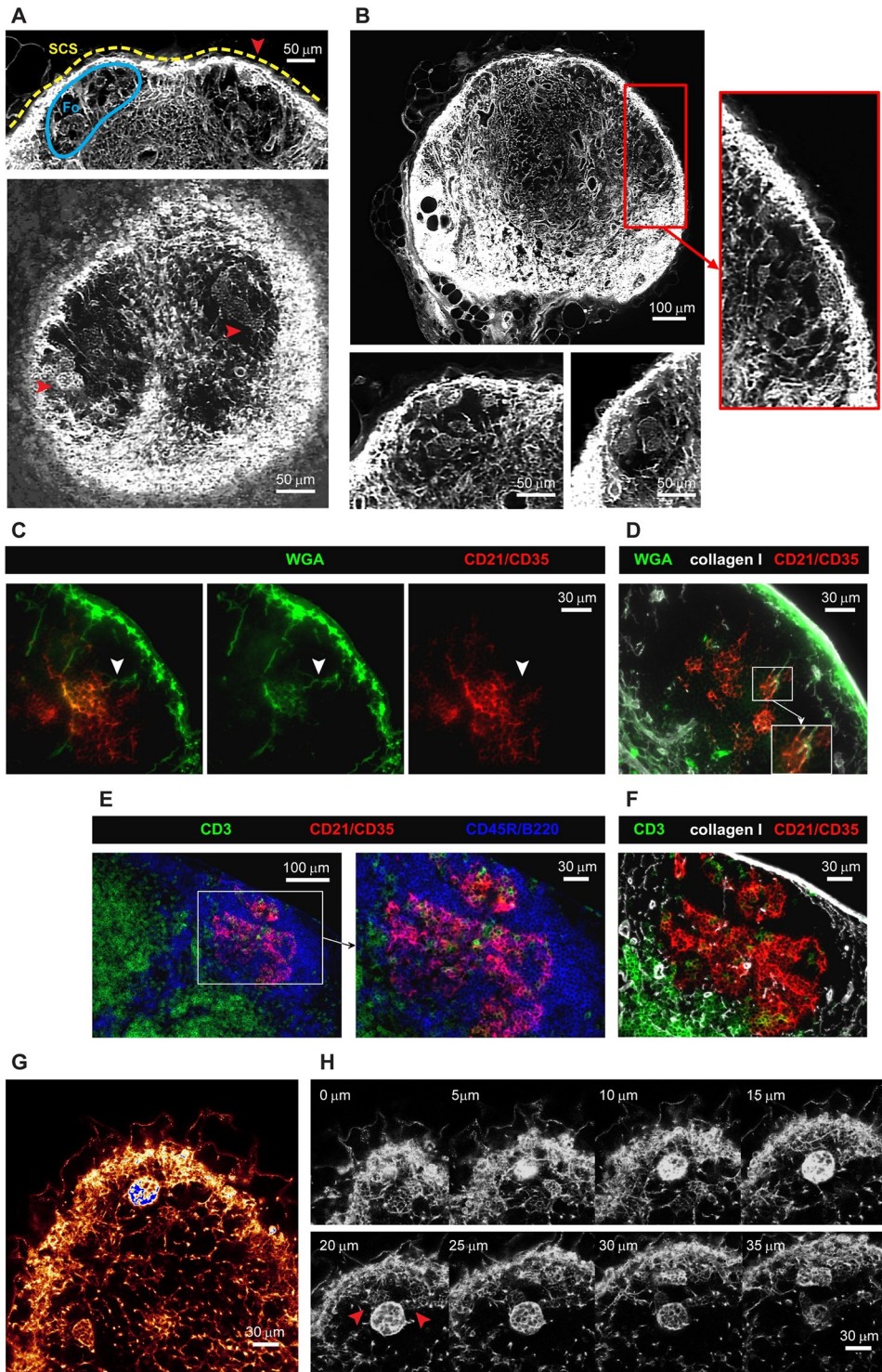

**Fig 4. The follicular conduit.** Three-dimensional EVIS images of a popliteal LN with WGA-labeled conduit paths contain brightly labeled multilobular spaces (red arrowheads) within the otherwise unstained B cell Fo underneath the SCS that can be repeatedly seen in 3D projections of 20 μm thickness (A) and 2D image slices (B). In immuno-labeled tissue sections of WGA-perfused inguinal LNs, WGA (green) is found in follicular conduits descending from the SCS (white arrowheads) connecting to cellular clusters expressing the FDC marker CD21/CD35 (C, D). The morphology and location of WGA-labeled cell clusters within B cell follicles (D) are generally consistent with the anatomy of FDCs in these regions (E, F) as co-staining with markers for B cells (CD45R/B220), T cells (CD3), and collagen I confirms.

High-resolution confocal images (with a voxel resolution of $0.36 \times 0.36 \times 1$ μm) of a WGA-perfused popliteal LN provide insights into the staining pattern within and around the WGA accumulations (G) and show a particularly bright cluster in several z steps (H) directly neighboring spaces with weaker labeling (red arrowheads). Images are representative of at least 6 LNs (from $N = 5$ mice) in which multilobular cell clusters could be observed. See also S4 and S5 Videos. CD,; EVIS, extended-volume imaging system; FDC, follicular dendritic cell; Fo, follicle; LN, lymph node; SCS, subcapsular sinus; WGA, wheat germ agglutinin.

therefore supports the concept that conduit segments descend from the SCS, branch through the paracortex, and richly supply perivascular sleeves, including those surrounding HEVs. Notably, these perivascular sleeves represent the first space encountered by cells exiting the bloodstream and suggest an underexplored role for the conduits in conveying molecules directly from the SCS to lymphocytes and antigen-presenting cells that have just entered the LN from the blood [18]. These data also lead us to conclude that these regions do not represent major sites of lymph drainage into the vasculature. Instead, we observed conduits frequently terminating in lymphatic sinuses that are blind-ended invaginations of the medullary sinuses, which is likely to provide the necessary outlet for accumulating lymph and potentially aids cell egress at these locations [49].

Analysis of over 280,000 conduit segments in the 3D topological model we generated revealed a homogeneous branching pattern, with bifurcations being the most prevalent branching structure and no more than 7 segments meeting at any one point. The level of connectivity that we measured between neighboring nodes is slightly lower than that measured by Novkovic and colleagues [33], who note the presence of highly connected nodes with more than 12 edges, based on confocal images of CCL19-expressing cells across a small fraction of the paracortex. The data presented here represent network parameters from an entire TCZ, providing a clear advantage to previous extrapolations from small LN regions. In contrast to their cell-based graph network, our conduit model provides a road map of the collagen-bearing conduit channels that the FRCs ensheath, including their exact lengths and orientation. It is possible that 2 or more conduit branch points in our model fall within the area of one cell body, to account for some of the differences in network topology, yet our data do not support the prevalence of highly connected nodes as they report. New opportunities are now likely to arise by combining the techniques we developed with those established for imaging FRC cell bodies, for example, to track the structure of entire conduit networks in response to immune stimuli or disturbance of FRC network integrity—phenomena that have only recently begun to be explored [10,13,14,33].

A striking feature that is visually obvious in our 3D images is the variation in conduit network density between the deep and superficial T-cell regions. The topology map of the TCZ conduits we generated allowed us to quantify significant differences in conduit segment density, segment length, and intersegment gap size between both zones. Our 3D imaging data are therefore consistent with several studies that previously identified a structural inhomogeneity within the TCZ in the LN paracortex. Whereas the deep zone has been described as loosely interspersed with a network of FRCs and conduits, the peripheral zone was noted to contain a much denser mesh and a higher abundance of HEVs [40]. The superficial TCZ [50] has also been referred to as the cortical ridge [40,51], or the peripheral T-cell region [52], and is continuous with the interfollicular regions between the B cell follicles closer to the SCS [53,54]. Although the biological significance of this structural segregation is still unclear, independent reports have pointed to an asymmetry in cell positioning in both zones. Naïve T cells tend to occupy the deep TCZ, whereas memory T cells preferentially locate to the superficial zones, and innate effector cells can often be found in the interfollicular regions [40,55,56]. Similarly, subtypes of resident and migratory DCs seem to preferentially locate to either the deep, the

superficial, the interfollicular zones, or regions close to the medulla [51,52,57]. It has also been frequently observed that following immune challenge T cells cluster in peripheral regions or locations close to the medulla [52,53,58–60]. It is therefore intriguing to note that Ki-67-expressing cells in the resting LNs we examined were often located in very close proximity to a conduit, and tended to localize to the periphery of the TCZs (S4 Fig). Interleukin 7 (IL-7) production may be higher in the peripheral TCZ [61], and close proximity to the FRC network might increase access to homeostatic survival and growth factors for memory or recently-primed T cells.

Our measurements of conduit density in the deep and superficial TCZs led us to conclude that although T cells within the superficial zone could remain in almost continuous contact with FRCs wrapped around the conduits, the larger gap size in the deep TCZ does not guarantee simultaneous contact for all T cells in this region. Although in vivo imaging studies suggested that T-cell motility is generally bound to the FRC network [45,62], T cells were observed to occasionally leave the FRC paths and migrate perpendicular to the FRC scaffold. Recent data established that T cells migrate in a sliding manner on the FRC network and suggest that fast scanning rates are achieved through low adhesiveness to the FRC substrate [45,63]. Our 3D data confirm that a dense continuous network is present to support the migration of cells across the TCZ, but the more open topology in the deep TCZ substantially increases the likelihood of an occasional loss of contact. Interestingly, theoretical studies of the FRC network have concluded that the odds of a cognate T/DC encounter are in fact not significantly increased by confining migration to a network [28–31].

When we used our 3D topology map of the conduit network as pathways to simulate FRC-bound T-cell migration, we observed an increase in the coefficient of motility as velocities increased across the range commonly measured in vivo [42,43], confirming that higher velocities translate to faster scanning rates. However, in these simulations, in which T-cell migration was solely restricted to the paths represented by the conduit network, we could not detect substantial differences in the coefficients of motility between the deep and superficial TCZs at any particular velocity. This may relate to the fact that although the density of the conduit networks differs in these 2 zones, their branching topology is very similar, with the network in the deep zone effectively representing a "stretched" version of that in the superficial zone. Some previous in vivo measurements recorded higher T-cell velocities in the deep TCZ compared with peripheral zones, which implies that motility coefficients could differ accordingly in these regions in vivo [46,64]. Future models of T-cell migration will now be able to incorporate our measurements of conduit network topology to model the conduit/FRC-guided component of T-cell motility, as we have shown here. However, accurate models will benefit from incorporating additional factors, including the effect of chemokinesis and chemotaxis [65] especially driven by CCL19 and CCL21 [3,59], the need for T cells to migrate around obstacles [29] including each other [66], and binding to DCs [46], as well as external factors such as confinement affecting the mode of cell migration [45,63].

In summary, our data provide quantitative support for the concept that the FRC and conduit network in the paracortex are arranged in a way that supports different processes in spatially distinct functional zones.

In the B cell follicles, conduits have previously been found similar in diameter and particle size exclusion (generally excluding large molecules of >70 kDa) to those in the TCZs, although they do not span the B cell follicle but descend as sparse short parallel channels from the SCS to converge with FDCs in the center of the follicle [21]. Here, we confirm this topology of the follicular conduits, but we also show that they supply a space surrounding the FDCs that we propose be termed "follicular reservoirs." These honeycombed spaces surrounding FDCs are remarkably well defined and can easily be distinguished in 3D images from the voids of

unstained cells surrounding them in the follicles. Although these structures could be easily identified in all PFA-fixed and LR white–embedded preparations of whole LNs, staining was frequently lost in acetone-fixed cryosections, suggesting that the fluorescent tracer is in soluble, unbound form within the follicular reservoirs. This phenomenon may also explain why these structures have not been seen in this clarity in previous studies. Our methods to label and preserve the material within the follicular reservoirs open the way for future studies to identify all the cell populations involved in their formation, to clarify the mechanisms by which molecules from the SCS accumulate within them, and to track the changes they undergo during immune activation and germinal center formation.

The identification of follicular reservoirs supplied directly by fluid from the SCS is important when considering the supply of antigen to B cells [21,67]. Although free diffusion limits the speed at which soluble antigen can reach the deeper follicular region from the SCS, the follicular conduit allows these materials to be rapidly channeled directly to B cells and FDCs in the center of the follicle, which have been shown to readily take up small noncomplexed molecules [68,69]. In the presence of local antibodies or small complement molecules soluble antigen could then be complexed and retained by FDCs to fulfill the prerequisite for sustained B cell activation [68,70]. We suggest that the follicular reservoirs we have identified are likely to play a pivotal role in this process. In addition, the follicular conduits may enable FDCs to access signaling molecules <70kDa delivered from incoming lymphatic fluid or cells near the SCS, in order to respond rapidly and directly to external stimuli.

It is important to note that the precise roles of the conduit system in distributing molecules to different LN compartments remain unclear. Several groups reported that DCs and B cells can obtain antigen directly from the conduits, providing a fast antigen delivery system that extends deeply within the LN [19,20,69,71]. However, Gerner and colleagues [52] have challenged this prevailing view, concluding that antigen dispersal to DCs and subsequent T-cell stimulation is dominated by conduit-independent diffusion. Instead, the conduit system may simply enable equilibration of fluid, with a subsidiary role in transporting signaling molecules [52]. Thierry and colleagues [22] recently provided additional support for the conduit system acting as a drainage system that allows IgM produced in the parenchyma to readily exit the LN and assure a rapid response to infection. Specifically localizing these different immune processes with respect to the spatial variations in the conduit network will help improve our ability to control and manipulate immune responses [52,72].

In summary, the data reported here present the first reality-based description of the conduit network across an entire murine LN paracortex. The extracted topology network provides a useful substrate for theoretical models of LN biology [38], such as 3D motility models and models of fluid distribution [32], as well as providing new insight into the structure of a network that is crucial to many immune functions.

## Materials and methods

### Ethics statement

All animal work was performed in accordance with the guidelines and the requirements of the New Zealand Animal Welfare Act (1999) and approved by the University of Auckland's Animal Ethics Committee (Approval number R965). During experimentation, mice were anaesthetized using 4% isoflurane (induction box) and then maintained on 2% isoflurane (40% $O_2$/air) via nose cone. At the end of the experiments, all mice were euthanized by cervical dislocation while under isoflurane anesthesia.

## Mice

C57BL/6J mice were purchased from The Jackson Laboratory (Sacramento, CA). Experimental protocols employ 9- to 22-week-old male C57BL/6J mice housed in the conventional animal facility unit at the School of Biological Sciences at the University of Auckland under environmentally controlled conditions (temperature and humidity) and a 12:12 h light/dark cycle. Animals were group-caged in transparent IVC cages with wood-chip bedding and environmental enrichment, in close proximity to other cages so that auditory, visual, and olfactory stimulation was present. We assessed animals daily for health and welfare and access to food and water.

## Tissue preparation for EVIS imaging

For in vivo staining of murine LNs, we used Alexa Fluor 488, 555, TMR, or 647 conjugated WGA, TMR-conjugated 2000 kDa dextran (Invitrogen, Thermo Fisher Scientific, Waltham, MA), and anti-LYVE-1 antibody (R&D Systems, Minneapolis, MN; clone 223322) that was fluorescently conjugated using the Alexa Fluor 488 Antibody Labeling Kit (Invitrogen, Thermo Fisher Scientific, Waltham, MA). For the labeling of LN conduit paths, fluorescently conjugated WGA was used as an anterograde tracer. After brief anesthesia, 50 μl of WGA-Alexa Fluor 488 (1 mg/mL) were injected into the footpad of C57BL/6 mice and let circulate for 30 to 60 min. This was followed by labeling the blood vascular system using a sequence of 1 mL fluorescent WGA-TMR (50 μg/mL at 20 μL/min) and 1 mL 2,000 kDa dextran-TMR (500 μg/mL)/ 2.5% gelatin mix (at 50 μL/min) in a post mortem local perfusion technique as described earlier [36]. Excised popliteal LNs were fixed in 4% PFA, 3% sucrose at 4 ˚C overnight before embedding in stable resin for EVIS imaging. Resin embedding was carried out by first dehydrating the tissue and infiltrating with LR white (hard grade, ProSciTech, Kirwan, Australia) followed by curing for 6 h at 60 ˚C as previously described [36]. The observable tissue shrinkage that occurs during this process was estimated to be 20%.

## Tissue staining and conventional confocal microscopy

To achieve triple staining of the blood vasculature, conduit channels, and lymphatic vessels in popliteal and inguinal LNs, briefly anaesthetized C57BL/6 mice were first injected with 50 μL anti-LYVE-1-Alexa Fluor 488 antibody (20 μg/mL, R&D Systems, Minneapolis, MN) into the rear right hock, an alternative injection site to the footpad, which is less invasive while allowing strong labeling [73]. After 8 h, 50 μL WGA-Alexa Fluor 647 (1 mg/mL, Invitrogen, Thermo Fisher Scientific, Waltham, MA) were injected in the same site, and after a circulation period of 1 h, the blood system of the whole body was labeled by injecting 100 μL of WGA-Alexa Fluor 555 (5 mg/mL, Invitrogen, Thermo Fisher Scientific, Waltham, MA) with 10 μL Heparin (100 units/mL) into the tail vein or vena cava of the anaesthetized mouse for a duration 2 min. Freshly excised murine tissue was fixed in 4% PFA, 3% sucrose at 4 ˚C overnight and embedded in LR white resin (medium grade, ProSciTech, Kirwan, Australia) for confocal imaging as described above. Standard confocal microscopy was performed using a Leica TCS SP2 equipped with a Leica HCX APO L 40.0 × 0.80 W UV water objective (Leica Microsystems, Wetzlar, Germany) at a voxel resolution of 0.36 × 0.36 × 1 μm.

## Immunohistochemistry

Freshly excised popliteal, inguinal, and mesenteric LNs were snap frozen in OCT compound (Sakura Finetek, Torrance, CA) and sectioned into 7 μm thick tissue sections. A protocol for multicolor immunohistochemistry established by Lloyd and colleagues [74] was adopted for

immunostaining, using up to 4 labels. These include antibodies against LYVE-1 (R&D Systems, Minneapolis, MN; clone 223322), collagen I (Abcam, Cambridge, United Kingdom), laminin (Abcam, Cambridge, United Kingdom), CD21/CD35-Biotin (Biolegend, San Diego, CA; clone 7E9), Ki-67 (Biolegend, San Diego, CA; clone 16A8), CD3e (BD Pharmingen, San Jose, CA; clone 500A2), and CD45R/B220 (BD Pharmingen, San Jose, CA; clone RA3-6B2). Primary antibodies were detected with Alexa Fluor 488, 555, or 647 conjugated goat secondary antibodies or Streptavidin (Invitrogen, Thermo Fisher Scientific, Waltham, MA) and nuclei labeled with DAPI (4',6-Diamidino-2-Phenylindole; Invitrogen, Thermo Fisher Scientific, Waltham, MA). Stained immunohistochemistry sections were mounted using ProLong Gold Antifade reagent (Invitrogen, Thermo Fisher Scientific, Waltham, MA) and photographed on a Nikon Eclipse Ni-U epifluorescence microscope (Nikon Instruments, Tokyo, Japan) using a SPOT Pursuit 1.4 MP monochrome camera (Scitech, Preston, Australia). Acquired images were pseudocolored, processed, and superimposed employing the Cytosketch software (Cyto-code Limited, Pukekohe, New Zealand).

### Ki-67 quantification

Analysis of the number of Ki-67+ nuclei per tissue region was performed in ImageJ (NIH, Bethestda, MD). Individual images from immune-labeled LNs were scaled and thresholded before extracting the area and the number of particles from the Ki-67 channel within regions of interest, covering either the superficial or deep TCZ based on the distribution of CD3e and laminin staining.

### EVIS imaging and image processing

EVIS is a confocal block-face imaging method that can capture large 3D regions of fluorescently labeled tissue up to several millimeters thick at a pixel resolution of up to 0.5 μm. In an iterative process, a resin-embedded sample is moved between a confocal laser scanning microscope (TCS 4D CLSM, Leica Microsystems, Wetzlar, Germany) and a precision miller (Leica SP2600 ultramill, Leica Microsystems, Wetzlar, Germany) both mounted on a high-precision three-axis translation stage (Aerotech, Pittsburgh, PA) and controlled by imaging software written in LabVIEW™ (National Instruments, Austin, TX), whereby previously imaged sections are removed in between imaging rounds as previously described [36]. Image acquisition was performed using an Omnichrome krypton/argon laser (Melles Griot, Rochester, NY) for sample illumination, a 20× water immersion lens (HC PL APO, 0.70 NA, Leica Microsystems, Wetzlar, Germany), 4× line averaging, and an image overlap of 50%. Individual 8-bit (gray-scale) images acquired at "1 μm pixel resolution" contained 512 × 512 pixels covering an area of 500 × 500 μm, providing a pixel resolution of 0.98 μm. By acquiring successive images at a z-spacing of 1 μm, an isotropic voxel size of 1 μm$^3$ was achieved. Under these conditions, the imaging throughput using a single color channel was calculated to cover a volume of 1 mm$^3$ at a voxel size of 1 μm$^3$ within 1 d (8 h). The dual-color image acquisition for the LN specimen presented here comprised 5 d in total. We are currently developing a new linear-scanning confocal imaging system, which acquires image data in parallel, with a speedup of around 100×.

Precise xyz-registration of the acquired image stack in conjunction with custom-designed image-processing and assembly software (LabVIEW™, [36,37]) enables the composition of seamless 3D images. As part of this process, individual images underwent background correction, deconvolution, and denoising before being merged into x-y mosaics and assembled into a 3D volume image. To further improve the quality of the generated 3D images and reduce the fluctuation of signal intensities between individual z planes across the 3D image stack, we employed an equalization algorithm to adjust the average image intensity in z

direction. Rather than having a fixed target for correction, a variable ("smoothed") ideal intensity was used for each z plane to account for the changing diameter across the spherical LN sample. Using the formula below, a correction factor $f(z)$ was obtained for each z plane and multiplied with the pixel intensities on the respective plane to create an equalized image. The corrected 3D image displayed a significantly reduced intensity variation and was better suited for image analysis.

$$f(z) = a * \frac{SA(z)}{A(z)} + (1 - a)$$

$f(z)$ = correction factor for each z plane
A = average intensity (above a fixed threshold to eliminate noise)
SA = smoothed average intensity
$a$ = 0.9 (to prevent over-correction)

## Network extraction and quantification

**Conduit network extraction.** The voxel-based 3D EVIS image of LN conduits was processed to extract a connected conduit network suitable for 3D measurements. This was performed using a modified set of the tools we previously designed to isolate the blood vessel network from fluorescent 3D images of a mesenteric LN [36]. The source code for these tools can be found at https://github.com/gibbogle/vessel-tools.git. In short, the grayscale 3D image is segmented using local thresholding, and the largest connected object selected to be skeletonized, followed by applying a tracing algorithm that transforms information from the segmented image and its skeleton into a topology map. This procedure generates a description of the network as a collection of connected tube segments, together with additional files allowing 3D visualization and manipulation. The image-processing parameters were chosen specifically to allow for capturing fine conduit channels within the TCZ, and medullary regions with a high intensity of staining were excluded. To further narrow down the selection of TCZ conduits, B cell regions near the surface of the LN were manually removed using the filament editor in Amira (Thermo Fisher Scientific, Waltham, MA).

**Exclusion of blood vessels surrounding conduit sleeves.** One feature of the conduit network inevitably created a challenge for the processing: large blood vessels are often completely surrounded by conduits, resulting in the formation of conduit sleeves, hollow tubes which cannot be reduced to a single centerline by the skeletonization algorithm (S3 Fig). Previous studies have manually excluded these parts of the network [33], but given the large size of the present network, we required a more automated approach in order to omit these sleeves. To this end, we utilized the blood vessel image data from the same specimen, optimized using our previously described tools [36]. We added them to the segmented image of the conduits and filled remaining gaps manually and by using the segmentation tool in Amira to obtain a combined 3D image of the conduits and blood vessels. As a result, conduit paths surrounding large blood vessels are reduced to the core blood vessel path, helping to avoid artifacts and preserving the continuity of the network. Alternatively, the segmented image of the blood vasculature was subjected to "region growing" in Amira and subtracted from the segmented image of the conduits, using a homemade tool that allows the addition and subtraction of pixel values between 2 images at the same location, in order to obtain a largely "blood vessel free" conduit image. Both conduit data sets, either containing filled blood vessels or no blood vessels, were subjected to network extraction and topology analysis separately. Depending on the experimental question, we used either of these networks for the subsequent analysis as described accordingly (Fig 2D–2F).

**Quantitative 3D measurements.** The network topology map obtained from the extraction allows direct read-outs of network parameters, providing the number of segments, their volumes, the number of vertices per branch point, the branching angles, and length measurements. We used the blood vessel–free conduit topology map in this measurement in order to obtain representative values for the conduit network without the contribution of blood vessels. In the calculation of branching angles, only segments with a length above 4 μm were chosen to avoid the jittering artifact cause by very short segments. Additional tools were utilized to calculate the minimum distance from points in the network to the closest conduit [36], providing a measure of spacing of neighboring segments and allowing the gaps between them to be visualized as lit voxels. In this calculation, all segments including potential blood vessels were assessed to avoid creating artificial gaps.

In order to investigate the possibility that there was a "preferred" orientation of conduit segments in the network, a method was developed to estimate the tendency of conduits to align with a set of 13 directions roughly spanning the 3D range. The directions were chosen corresponding to the lines connecting a point in a regular 3D grid to its 26 nearest neighbors. The distribution of segment directions over these 13 reference directions was calculated by summing the magnitude of segment projections onto the 13 lines, then normalizing. The results were visualized in LabVIEW (National Instruments, Austin, TX).

A "moving average" display providing insight into the relative segment density was obtained by first computing the averaged voxel densities of cubes with a set radius (e.g., 10 μm) while moving in 2 μm steps across the binary volume image, then rescaling the density values from 0 to 1 to 0 to 255, and finally visualizing the resulting averaged 3D image as a grayscale or false-colored (e.g., heat map) image using ImageJ (2D) and Amira (3D).

**Selection of subregions.** To specifically measure and compare anatomical differences between the outer and inner TCZs, spherical subregions with a diameter of 100 μm were selected for individual analysis from both zones. Because the identification of nontouching subregions within an irregular shaped 3D volume is not trivial and automated tools are lacking, we manually selected the center points for each of the subregions based on the observable segment density in z planes using ImageJ (NIH, Bethestda, MD). By specifying a center point and radius in a 3D cropping tool, these regions of interest could be isolated and their topology determined individually. As above, the topology map exclusive of blood vessels was used to obtain conduit-specific parameters, but blood vessels were included to estimate the distance distribution to the nearest conduit.

**Modeling T-cell motility.** Based on the current understanding that the FRC network provides a substrate for T-cell migration, we sought to simulate T-cell motility on the 3D conduit network. The coefficient of motility, $C_m$, which is analogous to a diffusion coefficient [75], was estimated by simulating the movement of a large number of cells on the network, subject to certain assumptions about speed and behavior at junctions. The procedure is as follows. A large number of cell paths through the network are simulated, the starting point (and starting direction) of each path chosen at random. Each cell is initially assigned a speed drawn from a Gaussian distribution with specified mean (here: 13 μm min$^{-1}$) and coefficient of variation − (SD) ÷ mean (here: 0.1). The cell moves with this constant speed along the network segments. When a segment junction is encountered, the branch taken by the cell is determined randomly, according to the following procedure. For each possible branch, $k$, the turning angle $\theta$ is determined, and for an angle less than 90˚, the probability weight $w(k)$ associated with that branch is computed as $\cos^4(\theta)$, the fourth power of the cosine of the turning angle, otherwise $w(k)$ is set to a very small value (0.001). The probability of taking branch $k$ is then given by $w(k)$ divided by the sum of all the weights. The actual branch taken is then determined in the usual way by generating a random variate with a uniform distribution. If a cell reaches a dead-

end in the network, the direction of movement along the segment is reversed. To reduce the encounter of dead-ends, which could skew the observed $C_m$, each tested network initially underwent a healing step of pruning and joining dead-end segments to neighboring vertices with a maximum branch length of 15 μm.

In short, if the junction-directed unit vector corresponding to the branch that the cell is on is $v(0)$, and there are $N_b$ branches the cell can take, with unit vectors $[v(k), k = 1, \ldots, N_b]$ directed away from the junction, then the turning angle onto the $k$th branch is given by the dot-product of 2 unit vectors ($\cos^{-1}$ is the inverse cosine function):

$$\theta(k) = \cos^{-1}(v(0) \cdot v(k)).$$

Then the probability of taking branch $k$ is given by

$$P(k) = \frac{w(k)}{\sum_{j=1}^{N_b} w(j)},$$

$$w(k) = (cos(\theta(k))^4 \quad \text{if } cos(\theta(k)) > 0$$
$$= 0.001 \ \text{otherwise}$$

The movement of each of 5,000 cells across the network was simulated in this way for a period of one hour. In order to avoid the possibility of a cell reaching the boundary of the network, the starting points were restricted to those falling within a sphere of radius 100 μm centered at the center of the LN, unless otherwise specified for TCZ subregions. The average squared distance of cells from their start points was computed at 5 min intervals and plotted. The estimate of the coefficient of motility ($C_m$) is the slope of the resulting line divided by 6. Because the curve is not a straight line, the average slope was estimated from the points at times 0 and 60 min (= T).

Let $(x_i(t), y_i(t), z_i(t))$ be the position of cell $i$ at time $t$, $N$ = number of cell paths simulated:

$$C_m = \frac{\sum_{i=1}^{N} \{(x_i(T) - x_i(0))^2 + (y_i(T) - y_i(0))^2 + (z_i(T) - z_i(0))^2\}}{6TN}.$$

To account for the tissue shrinkage that occurs during the embedding process (estimated to be 20% in each direction), a correction factor of 1.25 can be applied to all segment lengths in this simulation.

A selection of 20 tracks, normalized to the same starting point, was visualized in a spider plot using CMGUI. Calculation of $C_m$ was performed on the conduit network in which blood vessels were removed and on the full network including blood vessels, to account for the possibility that blood vessels can serve as additional migration paths.

**Visualization.** Three-dimensional images obtained from extended-volume or conventional confocal imaging were acquired as grayscale 3D tiff files in raw format and were pseudo-colored, processed, and superimposed with the 3D rendering programs Voxx (University of Indiana, Indianapolis, IN), ImageJ (NIH, Bethestda, MD), Amira (Thermo Fisher Scientific, Waltham, MA), or Imaris (Bitplane, Zürich, Switzerland). Visualization of 3D image data was performed by generating 2D projections of rendered volume images, isolating and displaying single z planes, or by accumulating several z planes over a range of 10 to 20 μm to provide "thick volume sections" that can allow better insight into the arrangement of fine structures over a restricted range of tissue. Selected programs such as Voxx and Imaris further allowed the generation of high quality movie files. The 3D rendering software CMGUI (Auckland Bio-engineering Institute, University of Auckland, Auckland, New Zealand) was employed for

 

rendering the network and the simulated cell paths. The distribution of segment directions were visualized in LabVIEW (National Instruments, Austin, TX). Graphs were generated in GraphPad Prism version 7.02 for Windows (GraphPad Software, San Diego, CA).

## Quantification and statistical analysis

All statistical analysis was performed in GraphPad Prism version 7.03 (GraphPad Software, San Diego, CA). Statistical parameters including the exact value of $N$, the definition of center, dispersion and precision measures (mean ± SD), and statistical significance are reported in Fig 3 and S4 Fig and the respective figure legends. Data is judged to be statistically significant when $p < 0.05$ by two-tailed Student $t$ test. In figures, asterisks denote statistical significance as calculated by Student $t$ test ($^*p < 0.05$; $^{**}p < 0.01$; $^{***}p < 0.001$; $^{****}p < 0.0001$; ns = not significant).

## Supporting information

**S1 Fig. Conduit channels focus on lymphatic sinuses.** Related to Fig 1. High-resolution confocal images of LN conduits, blood vessels (both co-labeled via perfusion in red), and lymphatic sinuses (LYVE-1, green) display conduits fusing into sinuses in close proximity to large blood vessels (arrowheads, panel A). In a moving average display (upper panel) generated from 3D LN images with WGA-labeled lymphatic channels that color codes pixel density in a rainbow spectrum, regions with a high density of conduits appear blue (B). These blue regions do not overlap with the location of blood vessels, seen as gaps in the conduit mesh (dashed circles), as can be taken from an overlay of the moving average image with the corresponding section of the conduit network image (B, middle). Instead, regions with high density of conduit stain show an association with lymphatic sinuses (arrowheads) that stain brightly with WGA (green) and lack a vascular core (red, B, lower). In multicolor fluorescent images of an immuno-labeled LN section, several collagen I+ conduits concentrate on a LYVE-1+ lymphatic sinus (C). LN, lymph node; LYVE-1, lymphatic vessel endothelial receptor 1; WGA, wheat germ agglutinin. (TIF)

**S2 Fig. The conduit network forms sleeves around blood vessels.** Related to Fig 2. Murine LNs were perfused with fluorescently tagged WGA and 2,000 kDa dextran to label the blood vasculature (red) and locally injected with fluorescently tagged WGA to label the lymphatic channels including conduit passageways (green). Multicolor fluorescent images of 2 μm LN sections show LN blood vessels are surrounded by sleeves continuous with the conduits (A). Close-up fluorescent images confirm the close juxtaposition of blood vessel endothelium (red) enclosed by a cell layer stained with lymph-borne WGA (green) against the background of autofluorescent cell bodies (blue, B). 3D reconstructed images of an LN volume image generated by EVIS imaging (at 1 μm pixel resolution, C) visualize the overall arrangement of the WGA-labeled channels including conduits and lymphatic vessels (red glow, left panel), the latter of which also stain positively for LYVE-1 (green, right panel), against the dense network of blood vessels weaving through the LN. Close-up images of 20 μm optical sections of a LN volume image illustrate how the conduit sleeves (red glow) fully enclose blood vessels (green, D). EVIS, extended-volume imaging system; LN, lymph node; LYVE-1, lymphatic vessel endothelial receptor 1; WGA, wheat germ agglutinin. (TIF)

**S3 Fig. Conduit network extraction produces unavoidable artifacts.** Related to Fig 2. A volume projection of the LN blood vasculature (blue) and conduit network (gold) demonstrates typical artifacts that occur during the segmentation of the fine conduit network around large

blood vessels (close-up box). Here, the conduit network encloses the blood vasculature entirely and forms large tubes or sleeves that cannot be interpreted by the skeletonization algorithm, resulting in the creation of many short segments along the conduit sleeve (red arrowheads), hindering realistic analysis of the network at these locations. LN, lymph node.
(TIF)

**S4 Fig. Proliferating T cells in LNs are located close to conduits and accumulate in the superficial TCZ.** Related to Fig 3. Multicolor fluorescent images of immuno-labeled LN sections reveal a close association between Ki-67+ cells and laminin+ conduits in the CD3+ TCZ of an inguinal LN (yellow arrowhead, A). Overview of CD3+ TCZs (green) in inguinal, popliteal, and mesenteric LNs in which Ki-67+ cells can often be found close to the border of the TCZ (yellow dashed line, B). The number of Ki-67+ nuclei is significantly increased in the superficial TCZ compared with the deep TCZ (C). Ki-67 quantification was performed in ImageJ based on data from 3 independent experiments (each point represents either the deep or superficial zone from a mesenteric LN, $N = 3$). Plots show mean ± SD; $^{**}p < 0.01$, Student $t$ test. Values for each data point can be found in S1 Data. LN, lymph node; TCZ, T-cell zone.
(TIF)

**S1 Video. Fly-through animation of an entire murine LN captured by EVIS imaging.** The 3D image reconstruction of this data set visualizes the lymphatic (red glow) and blood (green) passageways in a slice-by-slice view moving through z sections of 20 μm thickness and provides an interior view of LN subcompartments, including the staining-rich medulla, a dense mesh of conduit channels in the central TCZ, and the B cell follicles emerging near the SCS at the rim of the LN. Image reconstruction and animation was performed in Voxx. Related to Fig 1. EVIS, extended-volume imaging system; LN, lymph node; SCS, subcapsular sinus; TCZ, T-cell zone.
(MP4)

**S2 Video. Close-up animation of the LN paracortex.** This fly-through animation is moving slice-by-slice though the 3D volume image of a murine LN in optical z sections of 20 μm thickness, zoomed into the interface between the paracortex and the medulla. Blood vessels (green) penetrate through the LN volume rich in lymphatic staining (red glow), each surrounded by a conduit sleeve. Conduit channels frequently terminate on CS continuous with the lymphatic system (sinuses) of the medulla. The data set was acquired using EVIS imaging and visualized in Voxx. Related to Fig 1. CS, cortical sinuses; EVIS, extended-volume imaging system; LN, lymph node.
(MP4)

**S3 Video. The 3D surface representation and animation of the LN blood vessels, lymphatic sinuses, and conduit channels.** A crop from the edge of the paracortical region of a WGA-perfused murine LN exemplifies the tight relationship of the blood (red) and lymphatic passageways within the LN as conduit channels (gray) meet a plexus of lymphatic sinuses (green). Several blood vessels can be seen enclosed by a sleeve of conduits. The 3D image data were generated using EVIS imaging; lymphatic sinuses were isolated from the conduit data with the help of custom image-processing tools, and surface rendering and animation of the data was performed in Imaris. Related to Fig 2. EVIS, extended-volume imaging system; LN, lymph node; WGA, wheat germ agglutinin.
(MP4)

**S4 Video. The 3D reconstruction and animation of the blood vessel system and lymphatic channels of a murine LN.** In the first part of the animation the blood vasculature (red) is

shown in full and rotated around the y-axis followed by a slice-by-slice view of the LN moving through z sections of 10 μm thickness and displaying both the lymphatic (white) and blood vessel anatomy. Here, accumulations of the fluorescent tracer (WGA) used to visualize the lymphatic passageways (white) can be observed within the B cell follicles, which appear as spherical structures below the SCS, devoid of an organized conduit network. Within these locations, the tracer material is labeling interconnected clusters resembling the FDC network. The 3D image was acquired using EVIS imaging and reconstructed and animated in Imaris. Related to Fig 4A. EVIS, extended-volume imaging system; FDC, follicular dendritic cell; LN, lymph node; SCS, subcapsular sinus; WGA, wheat germ agglutinin.
(MP4)

**S5 Video. A close-up view of a "follicular reservoir" within a B cell follicle labeled with WGA within a murine LN.** A 3D reconstructed image of the conduit system (white) near the SCS is reduced slice-by-slice to open the view to a spherical cluster with strong fluorescent label, followed by building the image up again in a slice-by-slice manner. Standard confocal microscopy was performed with a voxel resolution of $0.36 \times 0.36 \times 1$ μm over a depth of 40 μm followed by image reconstruction and animation in Voxx. Related to Fig 4G and 4H. LN, lymph node; SCS, subcapsular sinus; WGA, wheat germ agglutinin.
(MP4)

**S1 Data. Numerical data and statistical tests used in Figs 2 and 3 and S4 Fig.**
(XLSX)

## Acknowledgments

The authors gratefully acknowledge the help and advice of Dr. Adrian Turner, Mrs. Amorita Petzer, Ms. Shorena Nachkebia, and the members of the Dunbar and LeGrice laboratories at the University of Auckland. We thank Ms. Jacqueline Ross and the Biomedical Imaging Research Unit (BIRU) at the University of Auckland for providing access to the image-processing software Amira. GB acknowledges the support of the Auckland Bioengineering Institute.

## Author Contributions

**Conceptualization:** Inken D. Kelch, Gib Bogle, P. Rod Dunbar.

**Data curation:** Inken D. Kelch, Gib Bogle, Gregory B. Sands.

**Formal analysis:** Inken D. Kelch, Gib Bogle.

**Funding acquisition:** P. Rod Dunbar.

**Investigation:** Inken D. Kelch, Anthony R. J. Phillips.

**Methodology:** Inken D. Kelch, Gib Bogle, Gregory B. Sands, Anthony R. J. Phillips.

**Project administration:** Inken D. Kelch, P. Rod Dunbar.

**Resources:** Gib Bogle, Gregory B. Sands, Anthony R. J. Phillips, Ian J. LeGrice, P. Rod Dunbar.

**Software:** Gib Bogle, Gregory B. Sands.

**Supervision:** Ian J. LeGrice, P. Rod Dunbar.

**Validation:** Inken D. Kelch, Gib Bogle, Gregory B. Sands.

**Visualization:** Inken D. Kelch, Gib Bogle, Gregory B. Sands.

**Writing – original draft:** Inken D. Kelch.

**Writing – review & editing:** Gib Bogle, P. Rod Dunbar.

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
