## [Editor Report · Decision Letter 0]

8 Aug 2019

Dear Dr Kelch, 

Thank you for submitting your manuscript entitled "High-resolution 3D imaging and topological mapping of the lymph node conduit system" for consideration as a Research Article by PLOS Biology.

Your manuscript has now been evaluated by the PLOS Biology editorial staff as well as by an academic editor with relevant expertise and I am writing to let you know that we would like to send your submission out for external peer review as a Methods and Resources paper.

*Please be aware that, due to the voluntary nature of our reviewers and academic editors, manuscripts may be subject to delays during the holiday season. Thank you for your patience.*

Please re-submit your manuscript within two working days, i.e. by Aug 10 2019 11:59PM.

Kind regards,

Di Jiang, PhD

Associate Editor

PLOS Biology

---

## [Decision Letter · Decision Letter 1]

13 Sep 2019

Dear Dr Kelch,

Thank you very much for submitting your manuscript "High-resolution 3D imaging and topological mapping of the lymph node conduit system" for consideration as a Methods and Resources by PLOS Biology. Your paper was evaluated by the PLOS Biology editors as well as by an Academic Editor with relevant expertise and by three independent reviewers. 

Based on the reviews, we will probably accept this manuscript for publication, providing that you will modify the manuscript according to the review recommendations and, particularly, that you will include the simulation requested by reviewer 3.

We expect to receive your revised manuscript within six weeks. You are welcome to submit your revised manuscript back before the deadline when it is ready. Your revisions should address the specific points made by each reviewer. In addition to the remaining revisions and before we will be able to formally accept your manuscript and consider it "in press", we also need to ensure that your article conforms to our guidelines; several of which are described below and marked with "***IMPORTANT: ". A member of our team will be in touch shortly with a set of requests. As we can't proceed until these requirements are met, your swift response will help prevent delays to publication.

Please note that you may have the opportunity to make the peer review history publicly available. The record will include editor decision letters (with reviews) and your responses to reviewer comments. If eligible, we will contact you to opt in or out.

Sincerely,

Di Jiang

PLOS Biology

ETHICS STATEMENT:

***IMPORTANT: Please add an Ethics Statements subsection in the beginning of the Methods section. The Ethics Statements in the submission form and Methods section of your manuscript should match verbatim. Please ensure that any changes are made to both versions.

-- Please include the full name of the IACUC/ethics committee that reviewed and approved the animal care and use protocol/permit/project license. ***IMPORTANT: Please also include an approval number if one was obtained.

-- Please include the specific national or international regulations/guidelines to which your animal care and use protocol adhered. Please note that institutional or accreditation organization guidelines (such as AAALAC) do not meet this requirement.

-- Please include information about the form of consent (written/oral) given for research involving human participants. All research involving human participants must have been approved by the authors' Institutional Review Board (IRB) or an equivalent committee, and all clinical investigation must have been conducted according to the principles expressed in the Declaration of Helsinki.

FINANCIAL DISCLOSURE:

***IMPORTANT: Please provide grant numbers if available in the submission form. 

DATA POLICY:

***IMPORTANT: Regardless of the method selected, please ensure that you provide the individual numerical values that underlie the summary data displayed in the following figure panels: Figures 2g-kmn, 3e-m, as they are essential for readers to assess your analysis and to reproduce it. ***IMPORTANT: Please also ensure that figure legends in your manuscript include information on where the underlying data can be found. You can write, e.g., "Values for each data point can be found in S1 Data. ".

Reviewer remarks:

Reviewer #1: In the accompanying manuscript, Kelchen et al. present a high-resolution global overview on the conduit system in a murine lymph node, and such an overview has thus far been lacking in the field. The image quality and the videos are compelling. The authors provide useful quantification of their data sets, including a modelling of T cell migration patterns and the identification of conduits draining to defined areas within B cell follicles. Altogether, the data represent a valuable resource for modelling of immune responses and deepen our knowledge on the conduit system.

There are only minor comments from my side.

1. Given the key role for EVIS in obtaining the images, the authors do not introduce this method in great detail, and one has to look in the Method section how the images were obtained.

2. Along this line, what is imaging throughput under the reported condition? For broader application of the technique, such information would be worth including in the manuscript.

3. How skewed is the distribution of Ki-67+ proliferating/recently proliferated cells (Figure S4)? This aspect has been overlooked in previous studies, so a quantification of the distribution would be of interest.

4. The accompanying videos would profit from a legend at the beginning such that one does not have to refer to the movie legend to assign fluorescent signals.

5. Figures 1h and i lack a scale bar.

Reviewer #2: The study presents an efficient methodology for an organ-wide 3D imaging and quantitative analysis of the structural features of the entire LN conduit system. It can be regarded as a natural continuation/extension of the Authors’ previous published approach to systematically characterize the blood vessel system in a murine network. A unique confocal block-face imaging system (EVIS) is used to generate a comprehensive set of 3D images of LN regions supplied by conduits. The imaging-data-driven approach in conjunction with some original algorithms for the processing of the voxel-based 3D EVIS image of LN conduits enabled the construction of the complete 3D topological model of the conduit system. The study identified a number of fundamental properties of the conduit network. These include the structural parameters, i.e., the node degree-, edge length- and branching angle distributions, the distance to the nearest conduits, the heterogeneity in network density within the T cell zone, as well as the features relevant for the functional role of the conduit system in LN (the existence of follicular reservoirs, computer simulation-based estimates of cell motility characteristics). 

The study opens new avenues for the development of spatially resolved computational models to examine quantitatively the role of the conduit system in LN fluid balance, transport of signaling molecules, and controlling the chemokine and cytokine fields affecting the lymphocyte motility and immune responses under various conditions. Overall, the results enable the quantification of the 3D conduit network parameters at exceptional resolution and precision. As such, they definitely will inspire further research in directions towards bioengineering of LNs, anatomically consistent mathematical modelling of the lymphatic system, and mechanistic understanding of the LN structure-function relationship. 

The manuscript is clearly written and well-structured. The supplementary information is sufficient to grasp the power of the study. 

I have no major concerns. Few minor comments: 

1 Lines 274-276. There is no continuity in presenting the parts of Figure 4: the text jumps from a,b, to g,h.

2. Starting from Line 606 onwards: Model of T cell motility. How robust are the predictions with respect to the value of the coefficient of variation? Recent data indicate that the variability could be much larger, i.e. up 50 ~ 0.5 (for example, Sivapatham S, Ficht X, Barreto de Albuquerque J, Page N, Merkler D and Stein JV (2019) Initial Viral Inoculum Determines Kinapse-and Synapse-Like T Cell Motility in Reactive Lymph Nodes. Front. Immunol. 10:2086. doi: 10.3389/fimmu.2019.02086). 

3. Line 618: Could one justify the choice of the functional form (i.e. the forth power of the cosine of the turning angle) be justified by reference to data or previous modelling work?

Reviewer #3: Here, Kelch et al perform painstaking, high quality imaging of the lymph node (LN) conduits following labelling with injected fluourescent proteins. The imaging is excellent and although most of the data only confirm what has been previously published on the LN conduits, this study provides information about the conduits across an entire LN, as opposed to single slices. The data will benefit future mathematical modeling and simulation of the LN microenvironment. 

The simulation of T cell migration present in this study adds an interesting element, although it is somewhat limited. The simulations are based on the premise that T cells remain in contact with the network of cells that form the conduit network. Most interesting is the difference in conduit density in the deep versus peripheral T cell zones. T cells migrating in the central T cell zone thus may not always contact the stromal/conduit network. The authors simulate T cell migration in both the deep and peripheral T cell zones, but the analysis assumes that the T cells contact the network to migrate. The findings from this simulation do not show any differences in migration of the T cells in either zone. However, it would be much better if the authors can simulate migration when the T cells do not need to migrate on the the network and compare the two zones. It seems to me that this may be more biologically relevant and the first time that this can be done because of the data generated in this study.

The authors also state a couple of times that the presence of extensive conduits in the B cell follicles was unexpected. Since others have published that conduits do serve the B cell follicles, and deliver soluble antigens to FDCs (ref 67 and others, as well as Nolte et al 10.1084/jem.20021801 in the spleen), these statements should be modified. The 3D imaging shown here does, however, improve our understanding of what these structures look like, in particular the FDC structures.

Minor comment: on line 196, page 12: simulated and not stimulated?

---

## [Editor Report · Decision Letter 2]

18 Nov 2019

Dear Dr Kelch,

On behalf of my colleagues and the Academic Editor, Burkhard Ludewig, I am pleased to inform you that we will be delighted to publish your Methods and Resources in PLOS Biology. 

Early Version

PRESS 

Kind regards,

Sofia Vickers

Senior Publications Assistant

PLOS Biology

On behalf of, 

Di Jiang,

Associate Editor

PLOS Biology